# Functionalization of TiO_2_ for Better Performance as Orthopedic Implants

**DOI:** 10.3390/ma15196868

**Published:** 2022-10-03

**Authors:** Sehrish Noreen, Engui Wang, Hongqing Feng, Zhou Li

**Affiliations:** School of Nanoscience and Technology, Beijing Institute of Nanoenergy and Nanosystems, University of Chinese Academy of Sciences, Beijing 100101, China

**Keywords:** TiO_2_, Ti implants, antibacterial properties, osteogenesis, functionalization

## Abstract

This review mainly focuses on the surface functionalization approaches of titanium dioxide (TiO_2_) to prevent bacterial infections and facilitate osteointegration simultaneously for titanium (Ti)-based orthopedic implants. Infection is one of the major causes of implant failure. Meanwhile, it is also critical for the bone-forming cells to integrate with the implant surface. TiO_2_ is the native oxide layer of Ti which has good biocompatibility as well as enriched physical, chemical, electronic, and photocatalytic properties. The formed nanostructures during fabrication and the enriched properties of TiO_2_ have enabled various functionalization methods to combat the micro-organisms and enhance the osteogenesis of Ti implants. This review encompasses the various modifications of TiO_2_ in aspects of topology, drug loading, and element incorporation, as well as the most recently developed electron transfer and electrical tuning approaches. Taken together, these approaches can endow Ti implants with better bactericidal and osteogenic abilities via the functionalization of TiO_2_.

## 1. Introduction

Titanium (Ti) and Ti alloy are currently the most widely used orthopedic materials. Their applications include micro-plates, micro-bone-screws, artificial bone joints, and fine surgical instruments. Ti-based materials have many excellent properties, including low density, high strength, corrosion resistance, good biocompatibility, magnetic compatibility, and no allergic reaction after implantation [1]. However, biomedical applications of Ti implants encounter a very important disadvantage; that is, Ti has no intrinsic antibacterial properties. Infections may occur after Ti implant surgery, resulting in implant failure, prolonged hospitalization, and increased cost to the health care system [2]. Microbial infections in the bones lead to a condition commonly known as osteomyelitis. A long-term and particular antibiotic medication is needed to cure it [3]. Production of implant material is a complex process [1] that requires safety standards in terms of biological and mechanical properties so that the health of patients may not be affected [4,5,6,7]. Ti and Ti alloys have advantages over Mg, Co, and stainless-steel alloys because Mg, as a degradable metal, cannot afford long-term service, Co has a high price, and stainless steel is not compatible with clinical magnetic resonance imaging (MRI) examination.

Titanium dioxide (TiO_2_) is the native oxide layer of Ti. Although TiO_2_ does not have intrinsic antibacterial properties, it brings many opportunities to improve the performance of Ti as a biomedical implant. In the process of oxide layer formation on the surface of Ti, various material treatment methods can be executed to obtain antibacterial properties. In addition, it is convenient to form TiO_2_ nanostructures that have better osteogenesis properties than Ti. In vivo, TiO_2_ may lead to better bone-implant contact by increasing cellular activity [8] and collagen type I expression [9]. Furthermore, TiO_2_ is one kind of semiconductor material with certain electronic and photocatalytic functions, which allows the development to find innovative methods to endow Ti implants with antibacterial properties and better osteogenesis ability.

In this review, we will discuss the various functionalization approaches of TiO_2_ to improve the performance of the Ti implant. The schematic diagram of this review is illustrated in Figure 1. The functionalization aims to endow the implant with antibacterial properties and enhance its ability to support osteogenesis. Apart from the well-established approaches such as topology, drug loading, and element incorporation, this review also discusses the most newly developed electron transfer and electrical tuning approaches. TiO_2_ is a material extensively involved in the electronic and photocatalytic fields for its ability to conduct electron separation and transfer. In recent years, it has been discovered that electron transfer between TiO_2_ and biological cells can also occur, and electrical tuning of TiO_2_ for the antibacterial properties is becoming possible. These novel approaches are worthy of more attention and research devotion.

## 2. Important Facts about Orthopedic Implant

### 2.1. Implant Failure

The success of bone surgical operations mainly depends on the quality of implantable biomaterials. Implant success is mainly halted by the infections caused by post-operative complications. Certain factors may lead to bacterial infections or even failure, including extensive damage to local tissues, improper fixation, smoking, diabetes, chemotherapy, irradiation, and inappropriate surgical techniques [10]. The implants may get an infection from surgery equipment, medical staff, room atmosphere, or bacteria in the patient’s blood. The outcome of these microbial infections sometimes becomes grave, leading to a second surgery, amputation, or even death [11]. Implant infections are mostly initiated by *Staphylococcus epidermidis* (*S. epidermidis*), *Staphylococcus aureus* (*S. aureus*), *Pseudomonas aeruginosa* (*P. aeruginosa*), and *Enterobacteriaceae* [12].

Implant failure may occur at early or late stage [13]. Lack of osseointegration may lead to early implant failure, whereas in late implant failures, osseointegration works well at the beginning but decreases later due to disease and biochemical overload [14]. Researchers have identified various reasons for implant failures, which include infectious and physical damage [15]. Implant failures can be minimized by maintaining hygienic measures, caring for physical damage, and regular review of implants.

Progressive bone loss occurs due to inflammatory lesions in the soft tissues associated with the implants [13] and peri-implant disease [16]. Poor hygienic measures, unmanaged diseases such as diabetes, and the use of corticosteroids in immune-compromised individuals may all lead to that situation [17,18]. Despite taking all necessary hygienic measures, bacterial infections may still occur. Studies have suggested that joint infections may take place in 1% of primary and 3–7% of multiple surgeries [19,20]. Patients with multiple surgeries have a higher risk of mortality and infection [19]. Implant infections and failures are a large economic burden on the health system. In the US, it costs more than $8.6 billion annually [19,21].

### 2.2. Fundamental Requirements of Orthopedic Implants

Bone is naturally composed of organic, inorganic, and collagen fibrils. The nano-hierarchical structures give shape and mechanical strength to bones [22]. The structures include small molecular amino acids forming tropocollagen helixes and nanoscale collagen fibers forming a microporous network of bones (Figure 2a). There is a crucial interaction between surface characteristics and the extracellular matrix for osteointegration [23]. Bone mesenchymal stem cells (BMSC) in the bone marrow are known to typically respond to metallic implants with the production of soft tissue rather than bone, which causes implants to fail [24,25]. Guiding stem cell differentiation to a desired specific line on the surface of the material is a key factor in the success of implants [26,27]. Osteoblasts are mature bone cells, whereas osteoprogenitor cells are pluripotent cells having the capacity to differentiate into different kinds of cells. Osteoblasts and osteoprogenitor cells are in direct contact with the implants.

For better outcomes, the hierarchical structures of the bone must be simulated by the implant with surface nanostructures to support bone tissue regeneration (Figure 2b,c). Apart from the surface nanostructures, other modifications, including nanoparticles, may help further. For example, bismuth oxide (Bi_2_O_3_) has features including electrochemical stability, high biocompatibility, and a medium band gap [28,29]. The contact of Bi_2_O_3_ nanoparticles and TiO_2_ nanocones resulted in a heterojunction that formed a built-in electric field and promoted the osteogenesis of BMSC on the basis of TiO_2_ nanostructures (Figure 2b) [30].

**Figure 2 materials-15-06868-f002:**
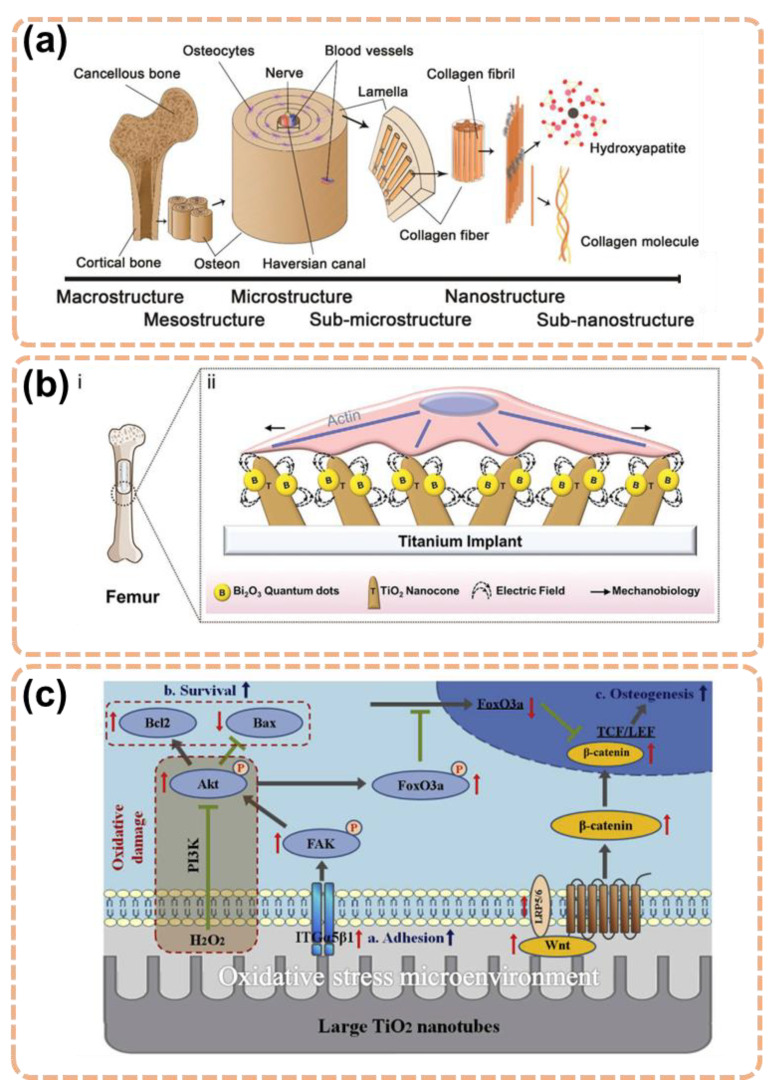
(**a**) The hierarchical structure of natural bone. Reprinted with permission from Ref. [31]. MDPI, 2022. (**b**) TiO_2_ nanocones with Bi_2_O_3_ quantum dots promotes osteogenesis. (**i**) Placing titanium implant in the bone defect; (**ii**) The electric field is built at the nanoscale interface of the implant. Reprinted with permission from Ref. [30]. John Wiley and Sons, 2021. (**c**) Osteogenesis pathways on TiO_2_ nanotubes in oxidative stress microenvironment brown arrow: enhancing the expression of a downstream gene; green arrow: inhibiting the expression of a downstream gene. Reprinted with permission from Ref. [32]. Elsevier, 2018.

## 3. Functionalization Approaches of TiO_2_ for Better Antibacterial and Osteogenesis Property

### 3.1. Topological Influence of the TiO_2_ Nanostructures

Topological modification is among the proposed methods to achieve surface functionalization. Studies have shown that surface nanostructure and topography may affect the migration, elongation, proliferation, and differentiation of stem cells [33,34,35]. In fact, cells and tissues in vivo will experience many topographic features ranging from nanoscale to microscale [36]. Thus, building a surface nanostructure on implants is an important research direction in the fields of artificial bones, joints, and dental implants [37,38,39]. The regulation of cell fate by surface topography is carried out by direct contact with adhering cells.

It has been widely accepted to form TiO_2_ nanotubes on Ti surfaces by doing anode oxidation (Figure 3a), and the annealing after anodization enhances the nanotube’s roughness and osseointegration capability [40,41]. Cell behavior is affected by the diameter of TiO_2_ nanotubes [40]. For instance, small nanotubes (30 nm in diameter) have been shown to promote BMSC adherence without significant differentiation, while larger nanotubes (70–100 nm in diameter) cause a dramatic lengthening of stem cells, which induces cytoskeletal stress and selective differentiation into osteoblast-like cells [42]. A diameter of 70 nm is the optimum size of TiO_2_ nanotubes for osteogenic differentiation of stem cells derived from human adiposity [43]. The diameters of TiO_2_ nanotubes are crucial for surface roughness and hydrophilicity. Several studies have shown that increasing diameter can increase antibacterial characteristics [44,45]. Ercan et al. found that nanotubes with a diameter of 80 nm had more antibacterial properties than the 30 nm diameter nanotubes against various strains of *S. aureus* due to higher hydrophobicity [46]. Other factors apart from the diameter, including the length, the gap between walls, and crystal forms, also influence the TiO_2_ nanotubes. Nano-engineered Ti prepared from hydrothermal etching has also been reported to be effective against gram-negative bacteria, *E. coli* [47].

TiO_2_ nanorod, another TiO_2_ nanostructure, also significantly influences the BMSC behavior [43]. The TiO_2_ nanorod array surface is very effective in regulating the differentiation of BMSC towards osteoblasts. In another study, TiO_2_ ceramics were synthesized and TiO_2_ nanorods were used to compare the BMSC cellular adhesion and self-renewal characteristics when commercial culture plates were used as the control group [48]. All samples demonstrated good biocompatibility from day 2 to day 8, suggesting that TiO_2_ ceramic promotes cell adhesion, renewal, and cellular morphology (Figure 3b).

Increasing the average surface roughness of the implant promotes osteointegration and is another topology-based surface modification [49]. The surface roughness enhances protein adsorption and osteoblastic functions [50]. The inorganic coating may include calcium phosphate/hydroxyapatite and certain peptides [51]. However, a thick layer of calcium phosphate coating has poor stability [52]. To address this issue, biomimetic strategies were devised, which have shown good versatility [49,53]. This coating has great osteoconductive potential in vivo [54].

### 3.2. Drug Loading and Release Based on the TiO_2_ Nanostructures

Antibiotics are very effective at killing bacteria, but antibiotics taken by oral or muscular injection have very low efficiency in treating infections in the bone. Localized drug release from the implant surface can solve the problem. TiO_2_ nanostructures such as nanotubes and nanopores are highly facilitated to do drug-loading [49,50]. TiO_2_ nanotubes are especially favored because of their larger surface area and one-end open feature [55]. The drug delivery of the nanotubes is significantly affected by the fabrication conditions. It is also found that drug release was promoted by increasing the dimensions (length, width, and diameter) of nanotubes [56]. Loading into the nanotubes with infection-reducing drugs, such as penicillin and streptomycin, largely improves the performance of titanium implants [57,58].

By increasing the dimensions of the nanotubes, drug release was promoted, but drug loss also increased during the rinsing process. To overcome this problem, periodic structures in the nanotubes are prevented, which demonstrated a significant improvement in the drug release control; the periodic structures largely reduced drug burst release from 77% to 50% and extended overall release from 4 days to more than 17 days [39].

The release control can also be improved by biodegradable layers (Figure 4a) [59]. Nanotubes can be coated with different layers of PLGA or CHI to improve drug release control and osteoblast adhesion [60,61]. Aw et al. enabled the release control of water-insoluble drugs by integrating TiO_2_ nanotubes with Pluronic F127 polymeric micelles and biopolymer chitosan coatings (Figure 4b). They reduced the drug release burst from 77% to 39% and extended the overall release from 9 days to more than 28 days [60]. These results suggest the great potential of a nanotube-based antibacterial system for sustained drug delivery to combat chronic infection and inflammation after surgery.

### 3.3. Element Incorporation

Apart from biotics, the antibacterial property can also be promoted by introducing antibacterial ions, such as silver (Ag), zinc (Zn), and magnesium (Mg) [62,63,64,65,66]. Jia et al. reported a method to incorporate Ag nanoparticles into TiO_2_ microporous coatings using polydopamine [62]. A sustained release of Ag^+^ ions for up to 28 days was observed, which endowed the Ti implant with long-term antibacterial ability. An additional trap-killing of the bacteria was enabled with these Ag nanoparticles (Figure 5a). Negatively charged bacteria were attracted toward the positively charged Ag nanoparticles and killed with more efficiency. More Ag doping to TiO_2_ for better antibacterial properties can be found in the literature [67,68,69].

Zn is an important trace element in the human body, and it has a pivotal role in DNA synthesis, enzymatic activities, biomineralization, hormonal activities, and antibacterial characteristics [70,71,72,73,74]. Zn doping in TiO_2_-based biomaterial has also been found to possess excellent antibacterial activities and better cell-material interactions [75,76]. The bacterial killing was due to the penetration of Zn^2+^ in the bacterial surface membranes [77].

**Figure 5 materials-15-06868-f005:**
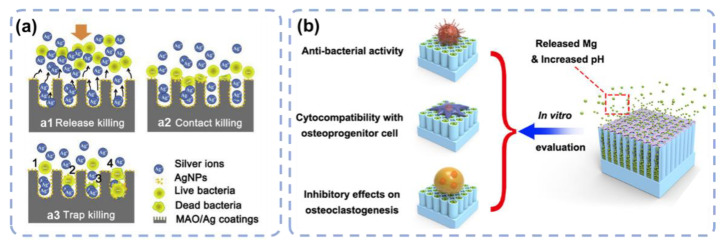
(**a**) The incorporation of Ag nanoparticles in TiO_2_ microporous structures to conduct bacteria killing. a1: Releasing of Ag nanoparticles to kill the bacteria; a2: Killing of bacteria upon contact with Ag nanoparticles at the TiO_2_ microporous structures; a3: Killing of the bacteria by trapping them in the microporous structures. Reprinted with permission from Ref. [62]. Elsevier, 2016. (**b**) The incorporation of Mg into TiO_2_ nanotubes to achieve both antibacterial and osteogenesis purposes. Reprinted with permission from Ref. [78]. American Chemical Society, 2019.

Mg is a microelement in the body and contributes to numerous cellular functions including enzymatic reactions, proteins, and nucleic acid synthesis; it is also effective in reducing inflammation and bone loss [79,80]. The incorporation of Mg can inhibit bacterial infection and osteolysis. Yang Y et al. designed a surface with Mg incorporated into the TiO_2_ nanotubes [78]. The surface demonstrated remarkable antibacterial properties, enhanced cytocompatibility, and inhibited osteoclast genesis, both *in vitro* and *in vivo*. The nanostructures and alkaline microenvironment during degradation were responsible for the antimicrobial ability. The continuous release of Mg^2+^ suppressed the osteolysis via down-regulation of NF-κB/NFATc1 signaling (Figure 5b). Mg doping has multiple therapeutic effects; however, an alkaline environment may pose a serious challenge in clinical use. Controlled release of Mg is the possible solution but needs further exploration [81]. Many other studies support that Mg incorporation can enhance the antibacterial and osteogenesis property of the implants [81,82].


### 3.4. Electron Transfer

In recent years, an antibacterial theory based on the electron transfer between the material surface and the microbes has been proposed. Electron transfer is a common event in the photochemical modulation of materials, as well as a fundamental event for the energy generation of organisms [83]. A group of microbes can do extracellular electron transfer spontaneously by transferring the electron outside the cells to environmental minerals [84]. However, using the electron transfer approach to inhibit implant infection is a quite new topic [85].

Vecitis et al. found that the antibacterial properties of single-arm carbon nanotubes are closely related to their electronic state. With the same diameter and length, metallic carbon nanotubes can cause severe deformation and collapse of the bacterial cells, while those in a semi-conductive state have no antibacterial properties [86]. Faria et al. found that the composite structure of Ag nanoparticles and graphene lamellae has a strong bactericidal ability, but graphene lamellae itself does not, suggesting that the electronic interactions between the substrate and the modified materials have a dominant impact on the antibacterial property [87].

TiO_2_ also has complex interactions with the bacteria and osteoblasts via electron transfer. TiO_2_ is a semiconductor, and biological cells can also be regarded as semiconductors [88]. Once contacted, they form heterojunctions, which may involve electron transfer. Therefore, functionalization based on the electron transfer property also influences the performance of TiO_2_ as an orthopedic implant. Au and Ag nanoparticles or graphene sheets deposited on the TiO_2_ surface can endow TiO_2_ with antibacterial properties [88,89,90,91,92,93]. On the Ag@TiO_2_ surface, electrons were stored on the Ag nanoparticles, and induced valence-band hole (h+) accumulation, which caused cytosolic content leakage of the bacteria (Figure 6a) [89]. On the Au@TiO_2_ surface, electron transfer was due to the plasmon effect of Au nanoparticles, which captured the electrons in the respiratory chain on the living bacterial cell membrane and transferred them to the TiO_2_ substrate. Au@TiO_2_ formed the Schottky barrier, which prevented the return of electrons, causing continued electron loss in the bacteria until death [91,93]. Similarly, graphene coating resulted in a large increase in the electrical conductivity of TiO_2_ because of the combination of the unpaired π electrons of graphene and the Ti atoms [94]. The enhanced electron transfer from the bacterial cell membrane to the graphene-TiO_2_ interface leads to bacterial death (Figure 6b).

Electron transfer also works for osteogenesis. Zhou et al. fabricated a SnO_2_–TiO_2_ heterojunction and hierarchical structure on the surface of the Ti implant [95]. The electron transfer among the hierarchical Schottky barrier significantly improved the osteogenic function of the cells around the implant both in vitro and in vivo (Figure 6c). In another work, they constructed a layered double hydroxide (LDHs)–TiO_2_ heterojunction, which promoted the transfer of holes in materials to the physiological environment, enhancing the antibacterial effect of the implant [96]. Ning et al. generated a microscale electrostatic field (MEF) by doing patterned NT (rutile) and IT (anatase) surface modifications on Ti [97]. The electron transfer between NT and IT zones formed a sustained built-in MEF, which polarized the BMSC and activated the expression of osteogenic genes (Figure 6d). The MEF greatly promoted bone regeneration around the implant.

Apart from TiO_2_, the Ti surface can also make electron transfer-based interactions with the bacteria. In a study by Wang et al., Ag was implanted on the Ti surface using plasma technology, and this modification changed the Ti surface from non-antibacterial to antibacterial [93]. The bacteria-killing was not due to Ag^+^ ion release, but due to the micro galvanic reaction at the nano interface between Ag nanoparticles and Ti substrate. The reaction disturbed the process of electron transfer in the bacteria respiratory chain and produced a large number of reactive oxygen species (ROS) in the bacterial cells, resulting in their death (Figure 6e).

**Figure 6 materials-15-06868-f006:**
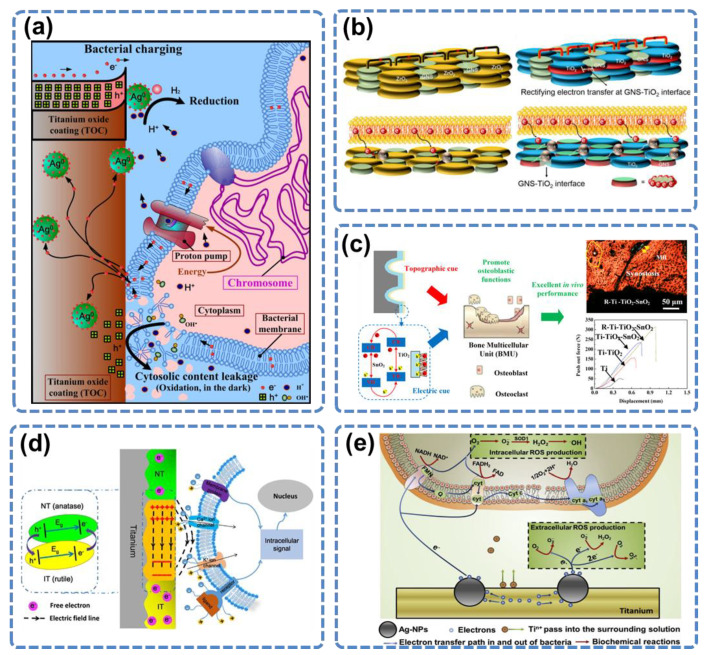
(**a**) Extracellular electron transfer induces bactericidal action of the Ag@TiO_2_ coating. Reprinted with permission from Ref. [89]. Elsevier, 2013. (**b**) The proposed mechanism explaining the antibacterial property of graphene nanosheets-TiO_2_ coatings is based on the electron transfer at the graphene-TiO_2_ interface. Reprinted with permission from Ref. [94]. Elsevier, 2020. (**c**) Illustration of the enhanced osteogenesis performance of titanium by an electric cue offered by the built-in electrical field of SnO_2_–TiO_2_. Topographic cue (red color arrow) and electric cue (blue color arrow) enhance the *in vivo* osteogenesis process (green color arrow). Reprinted with permission from Ref. [95]. American Chemical Society, 2018. (**d**) Demonstration of the mechanisms to generate MEF and the interactions between MEF and stem cells. Reprinted with permission from Ref. [97]. Springer Nature, 2016. (**e**) The electron transfer-based bacteria killing on the Ag@Ti surface. Reprinted with permission from Ref. [93]. Elsevier, 2017.

### 3.5. Electrical Functionalization

Based on the electron transfer mechanism of the above studies, researchers have further developed an innovative method to make the TiO_2_ surface obtain antibacterial properties through electrical tuning. In the beginning, it was found that an alternating current (AC) of about ±2 μA applied to the ZnO nanowires in a physiological solution could significantly improve the antibacterial property of ZnO after the current was removed (Figure 7a). The “sustained bacteria sterilization” was different from the “instant bacteria sterilization” because the latter was due to electroporation when AC was applied to the nanowires, but the former was due to surface functionalization by the electrical tuning [98]. After that, a 2 V low-voltage direct current (DC) power supply was used to conduct electrical treatment on the Ti plate with a TiO_2_ layer in the culture medium for 20 min. This DC tuning also changed the TiO_2_ surface from non-antibacterial to highly antibacterial [99]. After the electric tuning, TiO_2_ gained a strong ability to kill various bacteria and showed strong inhibition of biofilm formation. Meanwhile, the DC-tuned TiO_2_ surface had no negative effect on the osteoblast. The adhesion and proliferation of the cells were found to be as effective as those on the control TiO_2_ surface (Figure 7b).

## 4. Conclusions and Perspectives

In summary, Ti implants have a major concern regarding microbial infection, which may lead to implant failure. Various approaches have been carried out to deal with infections and promote osteointegration. TiO_2_ is not only the native oxide layer of the Ti biomedical implant but also a material widely studied in the photoelectronic and photocatalytic fields. Meanwhile, it is relatively easy to form nanostructures in the fabrication of TiO_2_ layers. Due to these properties, TiO_2_ has enabled various functionalization approaches to endow Ti with bactericidal and osteogenic abilities. In this review, we have discussed both the well-established and the newly-proposed approaches for TiO_2_ functionalization.

Topology, drug loading, and element incorporation are well-established approaches that have been developed for years. In some circumstances, the functionalization efforts may have conflicts. For the topological approaches, it is sometimes difficult to enhance the bactericidal and osteogenic properties at the same time because the topographies that can encourage BMSC and osteoblasts adhesion and proliferation will attract bacterial adhesion as well. In the ionic release approach, sometimes the released ions will harm the cells. However, innovations can still be made on the well-studied topics to overcome the problems they face. For example, nanoparticles can be used to help the TiO_2_ surface nanostructures, not from the aspect of topologies but by introducing a built-in electric field. Drugs can be incorporated into polymeric micelles, which may have more controllable behavior during the loading and releasing process. Metal ions such as Mg^2+^ and Zn^2+^ are found to have both bactericidal and osteo-enhancing properties; thus, incorporation of these elements into TiO_2_ via proper methods will be very helpful for Ti implants.

The electron transfer approaches are newly proposed in recent years. TiO_2_ is a material deeply involved in the electronic and photocatalytic fields for its ability to conduct electron separation and transfer. In the studies we discussed above, electron transfer between TiO_2_ and the biological cells also takes place and demonstrates advantages in balancing the two needs of the implant. Most of the electron transfer studies have declared that their approaches only have a killing effect on the bacteria but no adverse effect on the growth and differentiation of the osteoblasts. Based on electron transfer theory, the electrical tuning of TiO_2_ for anti-bacterial properties has also demonstrated success. However, the underlying mechanisms of the electron-based interactions of TiO_2_ with biological cells are far from explicit.

Most of the functionalization approaches are promising for clinical applications, as long as batch production can be realized and the production cost can be reduced. The electrical tuning method, however, needs extended evaluation because it involves *in vivo* electrical manipulation apart from material implantation. Whether the tuning parameters are safe for the body and their impacts on the surrounding biological systems are yet to be discovered. Of course, more animal experiments and clinical trials are needed for all approaches to translate them into human benefits.

In the future, it is welcome to conduct in-depth research on the bactericidal and osteogenesis mechanisms of the functionalization approaches to obtain a deeper understanding of the interactions among the implant surface, bacteria, and cells. This will enable researchers to design the functionalization methods more effectively and rationally and facilitate the clinical applications of Ti-based implants with more safety and convenience.

## Figures and Tables

**Figure 1 materials-15-06868-f001:**
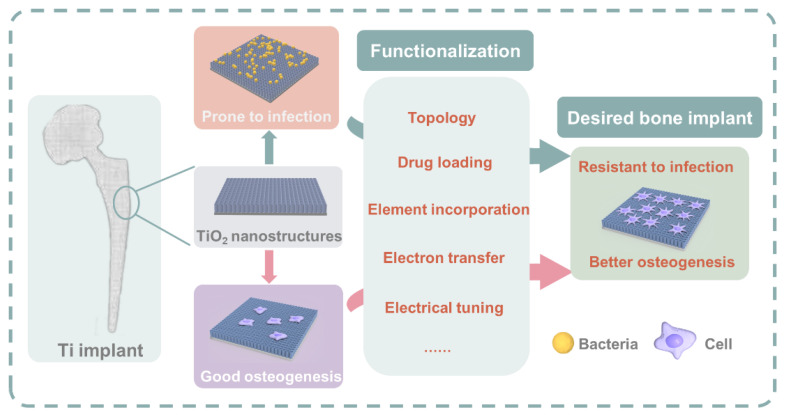
The schematic illustration of functionalization approaches towards the desired orthopedic implant.

**Figure 3 materials-15-06868-f003:**
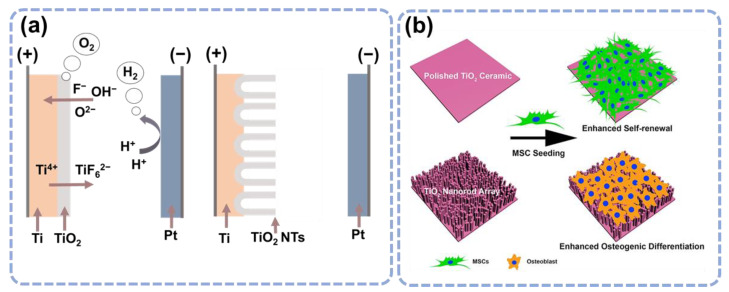
The topological nanostructures enhance osteogenic proliferation and differentiation. (**a**) The well-established fabrication process of TiO_2_ nanotubes. (**b**) The enhanced differentiation of BMSC on the nanorods. Reprinted with permission from Ref. [48]. John Wiley and Sons, 2016.

**Figure 4 materials-15-06868-f004:**
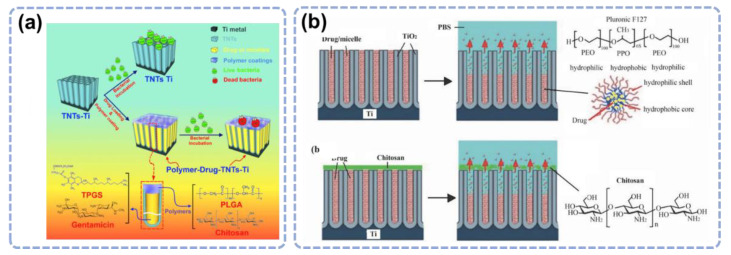
(**a**) Schematic illustration of the drug release control by TiO_2_ nanotubes. Reprinted with permission from Ref. [59]. Elsevier, 2015. (**b**) Polymeric micelles were used as nanocarriers and a chitosan polymer layer was coated on top of the nanotubes to control drug release. Reprinted with permission from Ref. [60]. Scientific Research, 2011.

**Figure 7 materials-15-06868-f007:**
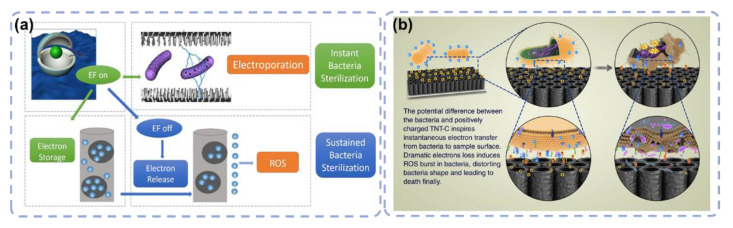
Electrical functionalization to endow the surface with antibacterial properties. (**a**) The first study to discover ZnO nanowires has gained sustained bacteria killing ability after AC tuning, which is different from electroporation when AC is applied to the nanowires. Reprinted with permission from Ref. [98]. Elsevier, 2017. (**b**) The systematic study to verify the electrical functionalization of TiO_2_ nanotubes by DC to endow them with antibacterial properties. Reprinted with permission from Ref. [99]. Springer Nature, 2018.

## Data Availability

Not applicable.

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
