# Peer review of "Functionalization of TiO2 for Better Performance as Orthopedic Implants"

_materials, 2022, doi:10.3390/ma15196868_

Round 1

Reviewer 1 Report

- From Figure 2, what is the specific role of the Bi2O3 NPs?
- Improve the resolution of Figure 3 and Fig. 6a
- Missing spaces in line 113 (formTiO2nanotubes), line 205 (as well asa), line 218 (TiO2also), and line 271 (20min)
- It is unclear which types of drugs can be helpful to incorporate in matrices where TiO2 is present. In this same sense, the conclusions should incorporate the perspectives and scope of this topic.

Reviewer 2 Report

This review paper is an exciting piece of work considering that it aims to discuss the various functionalization approaches of TiO2 to 43 improve the performance of Ti implant. However, minor corrections are required:

A deep revision of the English spelling and removal of Typos (L. 42, 113,205,218,240) will improve the readership.

The references in the text must be in the same bracket (L. 178,200)

The conclusion section should be rewritten, summarising the topics of the previous discussion.

 The perspective should be expanded and placed as section 3.6

Reviewer 3 Report

·       Add more references (2021-2022). You can add [1] New Titanium Alloys, Promising Materials for Medical Devices; [2] Cytocompatibility of pure metals and experimental binary titanium alloys for implant materials;

·        Show the novelty of the paper compared to the literature, however there is still much work on this topic.

·       Why you choose these materials? Add arguments beside cllasiccal alloys (Co alloys, Mg alloys and Stainless steel alloys)

·       In the Introduction section, the last paragraph should contain the scientific contribution and scientific hypotheses of your research. Complete, further elaborate the scientific contribution and scientific hypotheses of your research. Be explicit. In addition to the goal of the research (which was written), the novelty in the context of the scientific contribution should be pointed out. Scientific contributions should be written based on the shortcomings of previous research in the literature. In this way, the authors will better emphasize novelty and scientific soundness.

·       Analyze and discuss possibilities of practical application.

·       Add also some mechanical properties. Mechanical properties are also an important factor in choosing a biomaterial

·       Complete the conclusions with the limitations of the proposed methodology. Also write future research.

·       Generally, the quality of the writing could be improved.

Round 2

Reviewer 3 Report

The article was improved.